# Palliative Care and the Family Caregiver: Trading Mutual Pretense (Empathy) for a Sustained Gaze (Compassion)

**DOI:** 10.3390/bs7020019

**Published:** 2017-04-13

**Authors:** Joy Goldsmith, Sandra L. Ragan

**Affiliations:** 1Department of Communication, University of Memphis, 235 Art and Communication Building, Memphis, TN 38152-3150, USA; 2Department of Communication, University of Oklahoma, Norman, OK 73019, USA; sragan@gvtc.com

**Keywords:** palliative care, family caregiver, health literacy, communication training, compassion

## Abstract

In this conceptual piece, we survey the progress of palliative care communication and reflect back on a chapter we wrote a decade ago, which featured the communication concept of mutual pretense, first described by Glaser and Strauss (1965). This work will include an update on family caregivers and their role in cancer caregiving as well as a review of current palliative care communication curriculum available for providers. And finally, we will spotlight the conversation and research going forward on the subject of health literacy for all stakeholders; patients, families, providers, and systems. We feature one family’s story of incurable cancer and end of life to revisit the needs we identified ten years ago, which are still present. Goals for going forward in chronic and terminal illness are suggested in a health care context still too void of palliative care communication resources for providers, patients, and especially family caregivers.

*It had been nearly a month since I had seen her. When she opened the front door*, *I felt sure that she was dying. Sheila had moved into a house at the end of my street with her 24-year old daughter*, *Cass*, *and her daughter*’*s two young children*, *aged 1 and 8. The 8 year old was my daughter*’*s age and they had become fast friends and playmates. Sheila had raised her grandchildren while her daughter came and went as a very young single parent. Their arrival to the neighborhood had happened six months before*, *and upon my first meeting with Sheila*, *she shared about her ovarian cancer and how this had precipitated her move to our neighborhood—to be near another daughter who also lived a few houses away. In her seventh year with the disease*, *and still receiving treatment*, *it seemed clear that she had advanced terminal cancer. She mentioned things that told the story of her disease status: the recent discovery of a* ‘*new*’ *breast cancer*, *the language of hope confused by descriptions of ongoing aggressive treatments*, *and the notion of cure being mentioned as a possibility with her new doctors. Though I tried to share my thoughts about palliative care*, *this was never a topic we found a way to engage.*
*As I stood at her front door*, *the day I thought she looked like she was dying*, *she looked very thin. But her belly was enormous. Her eyes had a far off look and she described having bloating from diverticulitis*, *and that she was going to a GI doctor the next week to get it* ’*taken care of*’’. *I knew she was full of cancer and dying*; *I felt pretty sure that she might know this*; *and I was positive that no one in her world was talking with her about it.*

## 1. Palliative Care and Mutual Pretense

When we wrote a piece for Kevin Wright’s and Scott Moore’s *Applied Health Communication* in 2008 [1], we were novices in the scholarly investigation of palliative care. Differing significantly in our academic pathways, and one a mentor and one an advisee, we each had recently watched and endured the loss of a sister from terminal cancer. These losses coupled with our own academic experience and turn toward health communication ignited a call toward a newer kind of medicine called palliative care. Palliative care is unique in that it attends especially to relief from biomedical and psychosocial pain that accompanies serious illness [2]. This kind of care includes a team-based approach that attends not only to the patient, but also to the family caregivers by helping determine a patient’s goals of care through communication, attending to distressing symptoms, and coordinating care. Unlike the United States’ basic approach to hospice care, palliative care can be provided at the same time as curative treatments; it is appropriate at any age and at any stage of a serious illness [3]. In our first joint writing venture exploring this topic, we examined and found confusing, harmful, social scripts performed by physicians and patients/families at end of life, highly compelling. Looking back into the text we created ten years ago, we are reminded that our chapter was built around these ideas:
Americans are death-avoidant; thus we often die in hospitals, tethered to invasive medical machinery, and without benefit of having acknowledged death’s imminence and life’s remaining opportunities.As a result, we frequently live the end of our lives without the comfort of having addressed our emotional, psychic, spiritual, and existential pain—and without our family’s solace. Family members and patient alike are denied the relational closeness that living fully in the knowledge of an illness can bring.As patients and family members—we collude with our care providers to live in aggressive treatments as though we actually can avoid death, that our providers can solve the medical Riddle [4] of our illness. All parties to life-threatening or life-ending illness collaborate to produce a script boosted by the talk of treatment: we subscribe to Frank’s (1995) “restitution narrative” in that we believe that life, with appropriate (curative) medical treatment, can return to normal—to what it was before the anomaly of illness intervened, no matter the physical and emotional costs incurred [5].What Glaser & Straus describe is the ritual drama of mutual pretense, and it is enacted to manifest consensus that we are not dying [6]. It is an agreement between healthcare providers and patient/family, rarely spoken yet intricately coordinated, that all will behave as if advancing disease and loss of what was—is not an option [1].

In discussing this phenomenon of “mutual pretense” as described in these bullet points, we pointed up many of the problematic, undergirding props in current healthcare practice that augment its likelihood in the context of serious illness and dying:
The sender-based patient/physician model rather than the preferred collaborative model.The lack of patient-centered and family-centered care, such that the Voice of Medicine overtakes the patient’s Voice of the Lifeworld [7].The limited provider training in communicating with serious and terminal patients and their families.The ignorance about palliative care and hospice among providers, patients, family, community, and healthcare systems.The unavailability of palliative care in most U.S. hospitals.The belief by both medical caregivers and patients/families that hope inherently is cancelled if sustained and worsening illness is acknowledged.

We concluded the chapter by seeking antidotes for these concerns in patient-centered medicine, in narrative medicine (privileging the patient’s voice rather than trying to solve the Riddle), and in routinely incorporating palliative care into the care of patients with serious illness.

Now, ten years later, we wonder whether “mutual pretense” and this list of undergirding props are outdated notions, whether futile (allegedly curative) care is still prominent, whether palliative care is widely utilized, and whether our culture has become a bit more realistic about the inescapability of dying. This decade has been filled with research, initiatives, and training programs to better teach providers, systems, and caregiver/patient teams about the provisions of palliative care. We take this opportunity to pause and reflect on the productivity and labor of these years, and see what is ahead for palliative care and the family caregiver.

## 2. The Family Caregiver and Palliative Care Growth

The dimensions of burden, distress, and health of a family caregiver for a chronically or terminally ill loved one have moved into the cross-hairs of interest for researchers, hospital systems, providers, and health agencies. Health systems rely on the family caregiver to provide 80% of care that chronically and acutely-ill patients with cancer receive, and in managing adverse effects from disease and treatment in chronically and acutely-ill cancer patients [8]. Despite the demand for these high-level caregiving skills, family members often are unprepared and unable to navigate progressing disease and the decisions that accompany treatment processes [9]. Caregiver burden and distress are highest among those whose family member is living with increased symptom burden [10], underscoring the particular necessity for caregiver preparation and knowledge of symptom and pain management at home. As caregiver demands increase, caregivers who are unprepared may face greater adverse outcomes, including depression and mortality [11,12]. However, caregivers who report feeling more prepared for the caregiving experience report lower levels of personal strain during cancer care [9,11]. Cancer caregivers experience greater burden than non-cancer caregivers in the U.S., and provide more hours of caregiving per week, and cancer caregivers need significantly more help making end of life decisions than do non-cancer caregivers [13].

Despite the turn toward the family caregiver and the serious demands they endure in the course of disease progression or survivorship, caregiving distress and need are rarely monitored or even asked about in the clinical setting [14]. A current study of cancer family caregivers of patients with incurable cancer had significantly high levels of depression, and their anxiety was twice the level of the patient partner. Caregivers of incurable patients that described their efforts as curative in care reported higher levels of depression [15].

With the needs of patients and family caregivers still so significant, scholars across social science disciplines are now using the language of palliative care and end of life to identify their specialties. It is not unusual to see applicants for academic jobs in public health, medical anthropology, professional and technical writing, health informatics, and population science to present a focus in palliative care and chronic conditions. For us, the communication studies discipline, in particular, has cultivated scholars in this new area of research as well. The growing breadth and depth of providers and researchers in this area is also exhibited by the growth of palliative care services available in healthcare.

The Center to Advance Palliative Care (CAPC) published its 2015 assessment reflecting a slow but steady change in the number of hospital palliative care teams in the United States, though the standard and definition of what constitutes a ‘team’ in palliative care remains ill-defined. Sixty-seven percent of 50+ bed hospitals in the US report a palliative care team service. This is a 5% increase from 2011 and a 14% increase from the 2008 CAPC report. Another point of measure is the CAPC state-by-state grade card; an A grade represents more than 80% of hospitals within a state reporting palliative care teams [16]. The 2015 Report Card included an increase from 3% to 17% of states graded at an A. Notable is that the 2015 assessment included zero F’s for the first time (F = less than 20% of the state hospitals reporting palliative care teams) since the assessments began in 2008 [16].

We celebrate these advancements. But important gaps still remain. Despite the improving numbers of palliative care providers in care settings, still 1/3rd of 50+ bed hospitals do not claim any palliative care service, while another 1/3rd of the states received a grade of C (less than 60%) or D (less than 40%). The overall grade for the United States in 2015 was a B, unchanged from 2011 [16].

The National Hospice and Palliative Care Organization also tracks growth and change in the way we treat seriously ill and terminal patients in the US. In their assessment, agencies who participated in the study reported no change between 2006 and 2013 in the provision of formal pediatric palliative care services with specifically prepared staff nor the number of younger patients [17]. The odds of black and Hispanic children dying at home are far less than non-minority Americans indicating a lack of access to palliative and hospice care by this population [16].

These organizations demonstrate that where a person lives predicates his or her access to palliative care. The Northwest offers the greatest access to palliative care services. The south, the southwest, and non-urban care settings are far less likely to offer palliative care than urban locations [16]. The unfolding story of family loss without palliative or hospice care in my own neighborhood demonstrates the still-desperate need for this kind of care in the U. S.
*Emma was Shelia*’*s granddaughter*, *and my daughter*’*s playmate*. *Emma brought word that her MiMi (Sheila) had been inpatient since the day after I saw her at the front door*. *Her days at the hospital stretched on*, *one into the next*. *The moment she went inpatient*, *Sheila—mother*, *grandmother*, *and family bank*—*ceased being the primary parent for a one*, *eight*, *and twenty four year old*. *Her daughter*, *Cass*, *was in free-fall to accommodate both little children*. *She had relied almost exclusively on Shelia to care for her children while she worked as a server at a nearby restaurant*. *Without Shelia*, *Cass missed work and did not take her 3rd grader to school*. *Her parenting partner was no longer there*. *Shelia*’*s other daughter at the end of the street was less involved in her mom*’*s illness*, *and had two children of her own*. *Between these two daughters*, *neither of them visited their mom during her hospital stay*. *Emma began staying with us*. *She talked about the day her MiMi would get better*, *and come home*. *Now the family started to hemorrhage significantly*, *financially and emotionally*. *The house of cards had fallen*.

## 3. Palliative Care Communication Training

The National Consensus Project for Quality Palliative Care identifies the clinical practice of communication as central to palliative care, and calls for provider preparation for this particular clinical practice [18]. Healthcare governing bodies have also established communication competencies and education demands for current students and post graduate healthcare providers [16], and as such, a mounting number of training programs addressing communication in palliative care have been developed, marketed, revised, implemented, and sold.

Originating from the Communication discipline and grounded in communication theory, COMFORT Communication training has been offered regionally and nationally as a palliative care training for nurses, and palliative care team members since 2012. Unique to this training and the model, it is based on heavy integration of social science evidence, palliative care state of the science, and collaborative delivery crossing healthcare and academic disciplines [2,19]. The largest provider group receiving this training is nursing, followed by social work, medicine, and chaplaincy.

The Center to Advance Palliative Care (CAPC) began offering IPAL (Improving Palliative Care) in 2010—a central source of information for sharing expertise, evidence, tools and resources essential to the integration and improvement of palliative care in specific health care settings. IPAL has specific programming for EM (emergency), ICU (intensive care unit), and OP (outpatient). Only CAPC members can access resources, and the largest percentage of CAPC members is physicians. Relatedly, Palliative Care Leadership Centers selected by CAPC also model year-long mentoring for newly launched palliative care programs.

CAPC also supports VITAL Talk, a privatized communication-coaching model specific to patient and physician—as a training resource for communication in palliative care. Emerging from oncology, the coaches/trainers in this project rely heavily on physician experiences.

A simple Google search for palliative care communication training will produce institution-specific results for courses or certificates in aspects of palliative care at places like Stanford and Harvard medical centers, the University of Washington, Penn State, and Utah University, to name only a few. Countless resources are now available online that include podcasts, blogs, articles, and research exploring better pathways to get, use, and receive palliative care.

The installation of consult services and board certified healthcare providers widely defines what palliative care means to a healthcare system, while communication preparation and its impact on the quality of care lags behind in terms of strengthening processes and structures of those palliative care services [20,21]. Though we recognize the importance of training in palliative care and its impressive growth over the last decade, a shift in thinking about care and its quality involves still-needed compassionate action that moves beyond empathy, as well as attention to improved health literacy for patients, caregivers, communities, and care systems.

## 4. Health Literacy and Its Role in Quality Care

*For Shelia*, *the closest thing to a family caregiver was Cass—her daughter*, *with little children*, *few skills to survive*, *and now a sudden need to provide for all of them. The impact of her mom*’*s imminent dying accompanied by the silence around that reality created the perfect storm for her financial and emotional crash. Shelia remained in the hospital*, *and 8-year old Emma was lost and confused without her. Her life-long caregiver and bedmate had vanished. I stopped Cass in my yard as she dropped off Emma*, *and told her I thought Sheila was dying in the hospital.*

Health literacy includes personal characteristics and social resources needed to access, understand, appraise, and use information and services to participate in decisions relating to the health and care of the patient. A deficit model has led the research and development of measures and interventions in the study of health literacy—identifying the shortcomings of the patient and sometimes family caregiver and their ability to seek, find, understand, and use health information. Both the Institutes of Medicine (IOM) and World Health Organization (WHO) define health literacy as involving not only patient/caregiver level characteristics, such as cognitive and functional skills, but also healthcare system processes, structures, and providers [3]. In accord with the IOM, nursing and health literacy researchers [22,23], the synergy among those receiving care, providing care, and the resources provided by health systems converge to either fortify or dissolve health literacy barriers.

Health systems rely on family caregivers manage adverse effects from disease and treatment in chronically and acutely-ill cancer patients [8]. As patients become sicker, higher levels of health literacy are demanded of family caregivers. Despite the demand for high-level health knowledge and decision-making, family caregivers often are unprepared and unable to navigate progressing disease [9]. Caregiver burden and distress are highest among those whose patient is living with increased symptom burden [10].

Yuen and colleagues posit that cancer caregiver health literacy skills extend beyond understanding and accessing information and include the caregiver’s relationship and communication with the care recipient, relationships and communication with healthcare providers, communication within support systems, and managing the challenges of caregiving [23]. Models exploring oral health literacy [24], plain language use and understanding [25], as well as numeracy demands [26] are exciting additions to the growing number of applied examinations that are meant to improve the quality of care for patients and their family. Common among these new directions in health literacy is the awareness that conceptual understanding is not equivalent to word literacy; previous assumptions that linked the educational level of patients and caregivers with health literacy levels are no longer reliable [27,28].

Diane Meier, pioneer and leader in palliative care, has pointed at the health literacy challenges of providers and systems as key barriers in supplying high value care with simple solutions to the seriously ill [29]. Including all stakeholders (providers, communities, systems, patients, providers) in the health literacy problematic is an action that we believe will offer greater provision for improved care and quality of life. Palliative care is positioned to integrate health literacy improvement into its practice.

Health literacy that is informed and created by all who are participating in the acquisition, use, understanding, and communication about health decisions [22] may need to include the component of the relationships among participants to best be studied. Including the reality of the workload of health literacy to all participating, rather than measuring the deficits of those receiving care, promises to improve not only the tasks central to complex care, but more importantly the trust and commitment to improve outcomes across the board.

## 5. From Mutual Pretense (Empathy) to a Sustained Gaze (Compassion)

*Cass and I were in the driveway*. *A moment passed after I told her I thought her mom was dying. She leaned in and we hugged*. *She shuddered and whispered* “*I know*.”

The ability to take another’s perspective has long been identified as essential for patient/family centered care, which supersedes a biomedical or condition-guided focus to healthcare [30]. Concern for those who are suffering, whether the patient or the family of a patient, is the essence of empathy. Even though literature and anecdotal experiences plainly support the notion of empathy in healthcare training and practice, its translation into that training and practice remains unclear [31].

Empathic distress includes the feeling of immense care and concern, but also paralytic inaction; this kind of distress becomes focused on self. Distress, a product of empathy, can bring the focus back onto the provider, while “compassion leads to a focus on others” [32] (p. 201). Compassion is a feeling that emerges when facing suffering and being able to cope with that suffering (or to act in the midst of another’s suffering) [32]. The conflation of empathy and compassion in the literature, as well as in casual references in the context of healthcare, and specifically palliative care, has confused the role of each. Recent studies demonstrate that patients experience empathy and compassion very differently from providers. Empathy emerges as a first stage of compassion, but then is followed by a motivation to change that suffering—and this is a second and distinct response that differs from empathy [33]. Compassion moves beyond empathy and includes a process of reacting, acknowledging, understanding, seeking help, performing actions aimed at a solution, and even receiving personal satisfaction [33]. Compassion departs from the empathy’s response to acknowledge and understand suffering, and adds the distinct features of being motivated by action and seeking small, supererogatory acts of kindness [34].

So how might compassion serve this most challenging kind of care; palliative care? Are there ways that mutual pretense and empathy have partnered up unwittingly to deny patients and their families opportunities to act, to do, and engage the life that is present? Distress experienced by those around a dying person—whether family or provider—can paralyze if the fiction of cure and restoration is dramatized. This comes at the highest cost to the patient, who suffers alone in their false narrative of medical cure, and significant cost to the family who must live on knowing that better choices and more truthful communication could have changed the way a patient lived at the end of life.

Looking ahead to the next decade, we hope for increasing numbers of program announcements targeting palliative care research priorities, the family caregiver, and health literacy modeling. Rural and southern areas remain at a deficit for palliative care services, and the demand for programs and providers cannot keep pace. Innovative and integrated training models will be a pathway to educate providers, systems, as well as patients and caregivers about palliative care. Studies and implementations that include or are led by patient and caregiver stakeholders should become central to applied health communication and health literacy research.

As we wrote a decade ago, mutual pretense is eventually unsustainable. Today we do see progress in training, support for family caregivers, and for the dissemination of palliative care—all areas of concern in our early chapter. But the struggle of the patient and family, especially in non-urban settings, is all too resonant with our writings of 2008. Similar to my experience with Sheila and Cass, the mutual pretense collapses when conditions make its maintenance impossible. But the sustained gaze could be built on the actions of compassion; these actions move to acknowledge the reality of the patient’s biomedical and psychosocial state, understand the needs of this person in the context of the life, seek help to fulfill the outstanding needs for the patient and family, and perform actions that supply small and large solutions. This can be the alternative to the internal stare produced by mutual pretense. Gazing directly into the face of suffering, and opting for sustained presence instead of pretense, moves providers, caregivers, and patient into the truly cloaked comfort of compassion.

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
