# Peer review of "Palliative Care and the Family Caregiver: Trading Mutual Pretense (Empathy) for a Sustained Gaze (Compassion)"

_behavsci, 2017, doi:10.3390/bs7020019_

Round 1

Reviewer 1 Report

Thankyou for sending me this paper for review, I genuinely enjoyed it and I feel this would be true of other readers. I think I am happy to say that this article could go out without any changes, but I do have some minor comments that I think if addressed would improve the readabilty of this paper.

Firstly, I think your title is excellent and instantly grabs anyone who knows Glaser and Strauss' work. However, I did wonder if eluding abit to where you were taking your critque of Glaser much earlier on in the introduction would enhance it and make people more likely to read on in anticipation of the conclusion that you come to.

Next, as a UK based academic "hospice care" (mentioned on page 2) means something quite different. If the journal allows footnotes I may be tempted to give a sentence or two to define what hospice care means in the US setting; it certainly doesn't mean cessation of curative treatment in the UK and instead really just refers to a place rather than a system of care.

On page 3 in the bullet points the referencing system deviates into different forms. You use an author date, and superscript numbered system. This looked odd, and I couldn't work out why.

On page 4 on line 99 you say "where we are". I wasn't entirely clear who the "we" referred to. Was is US society, palliative care academic, the authors personally? Line 98 seems to suggest it is the authors themselves, but clarity may be desired.

On page 5 your discussion reminded me of an interesting piece of work from the UK about carer support. I'm not suggesting you reference this, but I thought you may be interested in Luker's et al's work full reference is "Development and Evaluation of an Intervention to Support Family Caregivers of People with Cancer to Provide Home-based Care at the End of Life: a Feasibility Study. European Journal of Oncology Nursing, 19(2), 154-161"

I generally greatly enjoyed the ethnographic type reflections offered throughout this piece. However, i thought the transition into and out of the one on page 6 was poor. I think a sentence or two extra was need to introduce it, even if this was abit tangential. The others were generally ok.

Finally, I really enjoyed pages 10 through 12. You produce some interesting reflections here and it rounds off the whole piece very well. Overall this is a well written paper which develops a timely critique of a seminal text.

Author Response

Dear Reviewer,

I am appreciative of the reviews I have received for my co-authored manuscript titled Palliative Care and the Family Caregiver: Trading Mutual Pretense (Empathy) for a Sustained Gaze (Compassion). Please find my reviewer responses here.

·      On line 54, I have added the descriptor of the US approach to hospice care, as the UK and other country approaches/definitions are quite disparate from the US approach.

·      I have removed the APA formatted reference within the bulleted list (line 69).

·      I have removed the language “see where we are” from line 99.

·      I appreciate the reviewer’s suggestion to learn about Luker et al.’s work addressing family caregivers. Thank you for this.

·      I have added a short introductory sentence to the reflection on page 6 to address the non-integrated read of this section.

Again, I offer my thanks to the reviewer for thoughtfully offering improvements to this writing effort.

Reviewer 2 Report

Excellently written manuscript that is thoughtful and reflective on the current state of palliative care in our country.  It flows logically and is interesting to read.  The case study is engaging.  My only suggestions are:

line 144 intro sentence needs to have word Pediatric since this is the focus of the paragraph

Author Response

I am very appreciative of the review offered here. 

In response to the request to include 'pediatric' in the opening sentence (line 144), I offer up that this paragraph is not focussed solely on the pediatric population. This paragraph also includes minority receipt of palliative care. With this in mind, I do include the word pediatric in line 147. 

With thanks---Joy